# A Scoping Review of Moral Stressors, Moral Distress and Moral Injury in Healthcare Workers during COVID-19

**DOI:** 10.3390/ijerph19031666

**Published:** 2022-02-01

**Authors:** Priya-Lena Riedel, Alexander Kreh, Vanessa Kulcar, Angela Lieber, Barbara Juen

**Affiliations:** 1Institute of Psychology, University of Innsbruck, Innrain, 52f, 6020 Innsbruck, Austria; Alexander.Kreh@uibk.ac.at (A.K.); Vanessa.Kulcar@uibk.ac.at (V.K.); Barbara.Juen@uibk.ac.at (B.J.); 2Department of Art and Design, University of the Arts Bremen, 28217 Bremen, Germany; Lieber.Angela@gmail.com

**Keywords:** healthcare workers, moral distress, moral injury, COVID-19

## Abstract

Ethical dilemmas for healthcare workers (HCWs) during pandemics highlight the centrality of moral stressors and moral distress (MD) as well as potentially morally injurious events (PMIEs) and moral injury (MI). These constructs offer a novel approach to understanding workplace stressors in healthcare settings, especially in the demanding times of COVID-19, but they so far lack clear identification of causes and consequences. A scoping review of moral stressors, moral distress, PMIEs, and MI of healthcare workers during COVID-19 was conducted using the databases Web of Science Core Collection and PsycINFO based on articles published up to October 2021. Studies were selected based on the following inclusion criteria: (1) the measurement of either moral stress, MD, PMIEs, or MI among HCWs; (2) original research using qualitative or quantitative methods; and (3) the availability of the peer-reviewed original article in English or German. The initial search revealed *n* = 149,394 studies from Web of Science and *n* = 34 studies from EBSCOhost. Nineteen studies were included in the review. Conditions representing moral stressors and PMIEs as well as MD and MI as their potential outcomes in healthcare contexts during COVID-19 are presented and discussed. Highlighting MD and MI in HCWs during COVID-19 brings attention to the need for conceptualizing the impact of moral stressors of any degree. Therefore, the development of a common, theoretically founded model of MD and MI is desirable.

## 1. Introduction

Morally stressful events, potentially morally injurious events (PMIEs), moral distress (MD), and moral injury (MI) have drawn scholarly focus in the wake of the current pandemic and previous pandemics due to lack of resources and changes in nursing practice [1,2,3]. In previous pandemics, having to choose between the ethics of professional duties and one’s individual fundamental values presented morally distressing situations [4]. Moral distress is a prominent phenomenon in healthcare professions [5,6,7,8,9] that has been studied across different aspects of healthcare provision, especially in nursing [5,6,8]. While there is not a unified definition of who constitutes a healthcare worker (HCW) across studies measuring MD and/or MI, other groups that have been studied in this context include physicians [10,11,12,13,14,15,16,17,18,19,20,21], mental health workers (e.g., psychologists and psychotherapists) [3,19,20,22,23], and chaplains [15]. In general, HCWs have been conceptualized as individuals who actively engage in the protection and the improvement of the health of society [24]. 

During the COVID-19 pandemic, HCWs have faced risks to patients’ lives as well as health risks to themselves. MD may be experienced when the ethics of patient-centered care and the requirements for protecting society come into conflict. For example, HCWs have been confronted with increased workloads and insufficient resources, such as time, in phases of rising infection rates. Furthermore, HCWs are confronted with fears of infecting themselves and family members [7,9,25,26]. Under conditions of frequently changing teams and lacking personal protective equipment (PPE), feelings of powerlessness in patient care may arise among HCWs [5,7,25,26]. Particularly in the context of a pandemic, the care of seriously ill patients, patients whose conditions deteriorate quickly, triage decisions, or the treatment of colleagues represent extreme stressors in the workplace [27]. This review seeks to collect triggers of MD and MI as well as differential impacts of moral stressors. Additionally, this review seeks to draw attention to vulnerability factors for MI and MD to occur in the context of COVID-19 among HCWs.

### 1.1. Moral Stressors

In order to differentiate the potential causal situations of moral distress (MD) and moral injury (MI), Litz and Kreig [28] propose a heuristic classification of moral stressors. According to this classification, events that have a direct or an indirect self-reference are conceptualized as moral stressors. In nursing literature, Epstein and colleagues [29] describe situations as morally distressing when they are characterized by a low degree of influence. This lack of control can lead to problematic or transgressive ethical action [29,30]. In this case, the individual feels compelled to act in a specific situation although the action enforces a moral transgression. Further, situations that trigger moral distress are characterized by a disregard for or an exclusion from relevant decision-making processes [29].

Potential moral stressors in healthcare settings are classified at the patient level, the team level, and the system level [29]. Moral stressors in healthcare settings have been evaluated in a variety of ways, predominantly across the occupational groups of nurses and physicians, working in general [5,6,9,31,32,33] or psychiatric [34] and oncologic [33,35] settings. In general, morally distressing circumstances in healthcare settings represent situations characterized by a lack of personal and professional competence [5,35]. Morally stressful situations of HCWs on the patient level have included futile (life-prolonging or risky) treatment [5,6,8], a lack of respect for patient autonomy [33], ignoring patient concerns [6,31,35], unethical behavior toward patients [5,31,35], and the involvement of relatives in caring decisions [9]. For nurses, not being able to deliver appropriate care, caring in unsafe conditions [6,35], and caring for critically ill or dying patients [32] were recorded as morally stressful situations. Among nurses, morally stressful events at the interpersonal team level further included witnessing unethical behavior by colleagues, working with colleagues in unsafe working conditions [6], a lack of respect for nurses’ autonomy [32,33], and a lack of involvement in nursing decision-making processes [8,32]. Studies on the triggers of moral distress in nurses have additionally identified institutional and organizational factors such as government or institutional policies and guidelines [6], the unavailability of medical staff [6], as well as perceived value incongruence of the organization [31].

In a military context, a terminology for major morally distressing events has been proposed [36,37,38]. The term potentially morally injurious event (PMIE) refers to a single abnormal and severe event that is largely outside of the individual’s control and has an extreme impact in the form of a threat to personal integrity or a loss of individual meaning-making elements [28]. PMIEs are rarely occurring events that take place within an individual’s life span [28,36]. Examples are the death of an individual belonging to a vulnerable group [39], leaving the dying behind [39], bringing harm to civilians or disrupting civilian life [40], and failing to prevent harm to others [40]. Considering this definition, PMIEs can also be referred to as morally traumatic stressors [41]. PMIEs can be based on one’s own behaviors and on the behaviors of others; for example, events based on one’s personal responsibility include the performance of a morally transgressive act. Witnessing a moral transgression without preventing it can also be experienced as a moral transgression [36,37,38]. On the other hand, witnessing a moral transgression can pose a PMIE [36,37,38] is especially significant in the case of authority figures as the betrayal of justice values by trusted authorities in high-stakes situations triggers MI [42].

### 1.2. Moral Distress and Moral Injury

Moral distress (MD) refers to the psychological experience of individuals in response to moral stressors [28]. Jameton [30] defines MD as the experience of psychological distress in situations where individuals are prevented from acting in ways they would have considered right based on personal values. Concurring, Corley [43] proposes MD as an experience that occurs as a consequence of an inability to execute a morally correct action due to institutional, social, or procedural restrictions, when the actor is aware of a morally correct action. Here, the failure to act morally relates to core values [43]. Universal core values include aspects of fairness, respect, caring, responsibility, and citizenship [44,45].

In the nursing literature, MD is referred to as a result of the violation of professional values and practices rather than personal values [29,46,47]. Central values in the healthcare profession have been articulated to include a commitment to excellence of practice, including accuracy in caring as well as individual and professional competence; a commitment to integrity and ethical practice; the maintenance of justice; and compassionate, respectful behavior toward patients and relatives [48,49]. Autonomy in decision making is another important professional value that has been established in nursing practice [49]. The experience of MD is related to environments characterized by a low possibility to affect change [29,30] due to disregard for or exclusion from relevant decision-making processes and a high demand to act [29].

According to Litz and Kreig [37], acute MD occurs rather often and the level of psychological, social, and spiritual distress that follows is moderate. The emotions felt in response to moral stressors are directly attributable and specifically related to immoral actions or their observation. Jameton [30] differentiates two forms of MD: (1) MD as an initial response represents the reaction of individuals in morally distressing situations, characterized by frustration and anger; and (2) MD in the form of reactive distress arises when individuals are not able to manage or resolve the morally stressful situation [30]. The associated emotions are conscious and stressful but they do not affect central aspects of self-perception. The effect of moral distress on psychological and social functioning is moderate and short-term [28]. Epstein and Hamric [50] propose the concept of moral residue. This arises when individuals are repeatedly exposed to morally stressful situations. It is assumed that moral distress accumulates as a consequence of each new morally stressful situation. People who are exposed to morally stressful situations do not fully recover from the distress they experience. This process is called the crescendo effect. Consequently, a moral residual arises after accumulation, in which the individual and the social environment are affected in such a way that moral integrity is threatened [46,50].

When MD persists and develops into a moral residue, or when PMIEs occur, the clinically relevant syndrome of moral injury (MI) can manifest in the form of a loss of trust in self, authority, and systems [42]. Litz and colleagues [36] studied the phenomenon of MI in veterans and defined MI as the potential consequence of PMIEs, manifesting at emotional, psychological, behavioral, and spiritual levels [38]. 

According to the working model of Litz and colleagues [38], social, individual, and environmental risks and protective factors influence the emergence of MI. The morally stressful situation is characterized by an inability of the individual to prevent the transgression. The committed or observed moral transgression cannot be classified into the personal world and self-concept and thus triggers a cognitive dissonance. In this phase, neuroticism and closeness to shame represent individual risk factors for both the extent of the cognitive conflict and the attribution of the immoral act as stable (unchangeable), global (overgeneralization of oneself as an immoral person), and internal to the failure of the person. Protective factors for MI at the individual level include a belief in a just world and self-worth. At the social level, proposed protective factors include a forgiving environment, which can prevent global and internal negative attributions. If the cognitive attribution of the event remains stable, internal, and global in the affected individual, the emotions of shame, guilt, and fear emerge. A consequence of shame can be social withdrawal. This social withdrawal can make corrective experiences by the environment more difficult. In the course of MI, individuals experience self-depreciation and an inability to forgive themselves. In this process, the inability to forgive oneself and self-depreciation, on the one hand, and the emotions of shame, guilt, and fear, on the other hand, perpetuate each other [36,51]. An aggravating circulation may emerge in which social withdrawal intensifies feelings of shame and guilt and these moral emotions lead to further social withdrawal. When experiencing MI, the moral emotions of shame and guilt are severe in their magnitude and they impact and cause chronic symptoms and problems with clinical relevance [28]. Symptoms of MI vary depending on the role of the affected person as witness or perpetrator of the morally transgressive act. When people see themselves as responsible for moral transgressions, the act becomes a dominant aspect of their self-definition. Transgressions based on one’s own responsibility relate to internalizing symptoms such as social isolation, anxiety, depression, or substance abuse [52,53,54]. Betrayal-based events lead to negative externalizing symptoms such as anger, loss of trust, and an inability to forgive others [37].

In addition to the perspective of Litz et al. [38], which focuses on the individual, Shay [42] highlights the social environmental component by defining MI as a betrayal by authority figures in high-stakes situations. This perspective focuses on the feeling of being let down by a legitimate social authority in the socio-cultural context [42]. The perception of betrayal can occur due to organizational or leadership malpractice, receiving immoral orders, or witnessing transgressive decisions [42]. 

Carey and Hodgson [55] draw on Jinkerson’s [56] definition of MI, which combines Shay’s [42] focus on authority and Litz’s [38] individually focused perspective, adding the component of spirituality and integrating MI into a bio-psycho-social-spiritual model. Referring to Carey and Hodgson [55] and Jinkerson [56], MI is defined as a particular trauma syndrome causing psychological, existential, behavioral, and interpersonal problems. The origin of this syndrome either lies in individual action or in witnessing and learning about transgressive acts by others that result in harm to others. These situations challenge the moral integrity of individuals, organizations, and communities. Additionally, transgressive actions or decisions of trusted actors holding legitimate authority can result in feelings of betrayal. Jinkerson [56] lists core and secondary symptoms that arise when experiences cause significant moral dissonance and remain unresolved. Core symptoms are: 


*“(a) shame, (b) guilt, (c) a loss of trust in self, others, and/or transcendental/ultimate beings, and (d) spiritual/existential conflict including an ontological loss of meaning in life. These core symptomatic features, influence the development of secondary indicators such as (a) depression, (b) anxiety, (c) anger, (d) re-experiencing the moral conflict, (e) social problems (e.g., social alienation), and (f) relationship issues (e.g., collegian, spousal, family), and ultimately (g) self-harm (i.e., self-sabotage, substance abuse, suicidal ideation, and death)”.*
([56], p. 126)

### 1.3. Vulnerability Indicators for Moral Distress and Moral Injury in Healthcare Workers

A broad empirical picture emerges regarding demographic and social predictors for experiencing moral distress and moral injury among HCWs. Regarding work experience, there are contradictory results. For example, in Shoorideh et al. [57] and Fruet et al. [33], higher age and longer work experience were associated with higher chances of experiencing moral stressors. In other studies, for example in Hamaideh [34], older and more experienced nurses reported fewer moral stressors. Among nurses, lower income [34] and higher educational levels were associated with higher frequency and greater intensity of stressors [33,34]. In terms of coping behaviors, using problem-solving strategies was associated with experiencing fewer moral stressors [58]. In nurses, maladaptive coping was associated with the experience of MD [59]. Nurses with high perceptions of comprehensibility, meaningfulness, and manageability were less likely to experience emotional exhaustion and depersonalization [60]. Additionally, low perceived empowerment and autonomy represented vulnerability factors for MD. Concerning professional attitudes, low work satisfaction and engagement were associated with experiencing MD [60,61]. 

At the social and organizational level, working under conditions of instrumental leadership—a leadership behavior focused on clear goals and fulfillment of tasks—was associated with experiencing MD [62]. Low staffing [8,63], increased workload [35], and restricted resources [5,8,32,35] represent further correlates of MD. Job demands were negatively associated with moral sensitivity and job stress correlated negatively with ethical climate [64]. Poor ethical climate was associated with distress in nurses [65]. Factors of ethical climate refer to the relationship between patients, managers, the hospital in general, and doctors; receiving help from the manager; involvement of doctors in decision-making; awareness of patients regarding what to expect from care; and friends who listen to work-related concerns [65,66]. In this context, lacking support from colleagues or supervisors [66]—represented as low professional [62] and social support [67]—posed vulnerability factors for MD [68]. Low informal and formal support may also be present in poor and unclear communication within the team [61] as well as in poor cooperation between different occupational groups of doctors, nurses, and students [5,61].

Vulnerability factors for experiencing MI among HCWs have not yet emerged from quantitative analysis in healthcare settings. Based on qualitative/theoretical considerations, potential risk factors for developing MI in response to PMIEs include lamenting the death of a vulnerable person and perceiving a lack of support from leadership, family, friends, or society [3]. Persistence of COVID-19 with further waves of infection and exposure to repetitive extreme moral stressors was mentioned as an additional risk factor [69,70]. Psychological unpreparedness to talk about extreme moral distress for societal or community reasons has also been proposed as a vulnerability factor [3]. Possible consequences of MI are burnout or job abandonment [71].

### 1.4. Scope of This Paper

We observe a research gap concerning predictors of MI and definitions of moral stressors, as well as in the identification of potentially morally injurious events in healthcare workers. Furthermore, we consider the present COVID-19 pandemic as a situation that makes it difficult to differentiate between normal moral stressors and potentially morally injurious events. In addition, the duration and omnipresence of the pandemic may lead to more severe consequences and exacerbate previously identified predictors. Thus, the objectives of the current review are: i.The identification of moral stressors, PMIEs, MD, and MI in HCWs during COVID-19.ii.The identification of predictors of MD and MI in HCWs during COVID-19.

## 2. Materials and Methods

### 2.1. Literature Research

An integrative literature review following the PRISMA statement [72] was conducted to identify moral stressors, PMIEs, MD, and MI in HCWs. The review protocol according to the PRISMA extension for scoping reviews (PRISMA-ScR; [73]) is available in the Appendix A (Appendix A. PRISMA-ScR Checklist).

In the systematic literature search, the first author reviewed the databases of Web of Science Core Collection (WoS) and PsycINFO via EBSCOhost up to 1 October 2021. The authors recognized as HCWs (a) health service providers delivering personal or non-personal services, including health professionals, health associate professionals, nursing and midwifery associate professionals, traditional medicine practitioners, and faith healers (including chaplains/clergy); and (b) health management and support workers including administrative staff, management, and accountants [24]. The literature research was conducted using the block search strategy [74] and included the following keywords: moral distress OR moral injury AND healthcare workers OR healthcare professional OR healthcare provider OR healthcare personnel OR doctor OR nurse AND COVID-19 OR coronavirus OR 2019-nCoV OR SARS-CoV-2 OR COV-19. The publication dates of the studies were filtered to 2020–2022 (early access). Limiting the studies to this period aims to directly compare the constructs through a common pandemic context. In addition, an automatic sorting of the titles by relevance took place on Web of Science.

### 2.2. Identification

The inclusion criteria for this study were: (1) measurement of either moral stress, moral distress, PMIEs, or MI among health workers; (2) original research using qualitative or quantitative methods; and (3) availability of the peer-reviewed original article in English or German.

Exclusion criteria were: (1) measurement of general psychological distress or a focus on treatment and response to moral distress, addressing strategies to mitigate the influence of moral stressors; (2) collection of the constructs from target groups other than HCWs; and (3) a lack of stringency in the application of the methodology, characterized by qualitative studies in which ethical approvals were not described and quantitative studies in which instruments were misapplied.

### 2.3. Screening and Selection

Initially, the titles and abstracts of potentially relevant studies were screened for eligibility. Articles that could not be accessed were excluded. Subsequently, the full texts were checked for thematic relevance and methodological quality. Suitable studies were integrated into the review. The review was organized according to existing theoretical proposals [30,36,42] as well as based on the consideration of the classification of stressors in nurses [29]. Theoretical proposals include the conceptualizations of MD [29,30] and MI [36,42,56]. Significant information on the constructs of MD, PMIEs, and MI was noted and subsequently synthesized. 

## 3. Results

### 3.1. Study Selection

The initial search yielded *n* = 149,394 articles from Web of Science and *n* = 34 from EBSCOhost. By using a filter, *n* = 22,097 editorials, reviews, and opinions were excluded. Automated sorting in Web of Science by relevance allowed for the exclusion of *n* = 126,626 articles that did not include the terms of interest. Three duplicates were excluded from the analysis. After reading the titles and abstracts, a further 31 were excluded, due to including neither MD, MI, nor PMIEs. One article was not available and thus excluded. Thirty-three articles were analyzed as full texts. Of these, nine articles were excluded because they violated inclusion criteria in terms of content or methodology. Four opinions and essays were also excluded. Nineteen articles were included in the integrative analysis. Figure 1 illustrates the study selection process.

### 3.2. General Characteristics of Studies

Table 1 presents a summary of the results from the included studies. Of the 19 studies, 6 studies were conducted in the United States, 3 studies were conducted in England, 2 studies were conducted in China and the Netherlands, and 1 study each was conducted in Romania, Norway, Israel, Australia, and in an Italian-Austrian collaboration. One survey was designed across six countries. Eight studies used a qualitative design. Six studies used a quantitative design with one measurement point, three studies used a quantitative design with multiple measurement points, and two studies were validation studies.

### 3.3. Synthesis of Results

Three main areas were identified. The area of “moral stressors during COVID-19” subsumes a broad array of morally distressing situations and circumstances (causes for moral distress) referred to by the reviewed studies. Additionally, this section includes PMIEs identified by quantitative studies using an adapted version of the Moral Injury Events Scale (MIES [77]) or described as causes for MI in qualitative studies. The second area includes the consequences of moral stress on individuals: “MD and MI during COVID-19” subsumes studies that measure the frequency and/or intensity of experienced moral stressors; studies using the Moral Injury Symptoms Scale-Health Professional (MISS-HP; [11]); and qualitative studies reporting on MI. The third area includes studies that measured “vulnerability factors of MD and MI during COVID-19.” 

#### 3.3.1. Moral Stressors during COVID-19

Moral stressors for HCWs during COVID-19 originated at the level of patient care, interpersonal relationships, and at the organizational level.

Patient-related moral stressors referred to the conflict between patients’ interests and caregivers’ safety, the conflict between the priority of protecting patients’ lives and the goal of delivering usual and appropriate care [9], witnessing inadequate provision of care [7,23], and the conflict between obligatory isolation of patients and patients’ freedom [9,23]. Additionally the fear of abandoning colleagues in the wake of their own infection represented a moral stressor [13]. In this context two studies reported on the experience of PMIEs [18,79]. In an Israeli survey, one’s own transgression of moral values was reported at 32%, 46% felt they witnessed things that were morally wrong, and 49% reported having experienced at least one transgression by others [79]. During the COVID-19 pandemic, the neglection of ethics of care was present in different degrees and magnitudes. For mental health HCWs wearing personal protective equipment represented a moral stressor [13] hindering confidence building due to limited visibility of facial expressions [23]. Other moral stressors included priority setting dilemmas [14] and restrictions on visitation rights, especially concerning dying patients [9,13]. Caring for patients without family contact and accompanying dying patients in the absence of family or spiritual support were stated as moderate moral stressors [25]. In this context, new role responsibilities [22] or loss of professional distance [22] emerged for HCWs. Mental health HCWs reported blurred roles when colleagues became clients [23]. In a study from the Netherlands, the most evident moral stressor was insufficient emotional support for patients and relatives [16]. Experiences of stress for all groups of HCWs included the inability to provide emotional support to patients when they or their relatives were anxious and stressed as well as the inability to provide a dignified death for the patient’s relatives [16].

In interpersonal relationships, moral stressors existed in doctor-nurse and nurse-nurse relationships in the wake of the uncertain pandemic situation, as well as in the nurse-patient relationship when cultural differences and communication difficulties were present [9]. Interpersonal stressors were concretized in terms of changing teams, leading to interpersonal conflict [7] and working with colleagues lacking professional competence in critical care [7,12]. Barriers in collaboration with physicians were described as differing views in treatment planning, disregarding nurses in relation to patient treatment decisions, and conflicts with relatives in relation to the use of scarce resources [7]. Interpersonal work-related concerns were present when working with colleagues who were not following safety guidelines or who were acting unsafely [12].

Moral stressors due to organizational constraints were identified in the form of conflict between scarce resources and equal distribution [7,13], hindered care due to a lack of financial support [12], resources of time [12], or staff [13,16], as well as damaged [22] or lacking protective equipment [13,14,25]. Exposure to unsafe working conditions can also be described in terms of PMIEs: in one study, betrayal by hospital leadership and by others was reported in 55% and 62% of respondents, respectively [79]. Experiences of betrayal by management during COVID-19 were concretized by lacking management support, perceiving treatment during the pandemic as dehumanizing, and being treated as a replaceable resource. Employees reported a lack of empathy, appreciation, and respect from supervisors [19]. In the COVID-19 pandemic, organizational stressors were apparent in the form of low organizational support [1] and in conflicts between ethical principles and ethical decision-making [9,13]. The organizational barriers were characterized by directives such as care during crisis conditions [7] with increased patient volume, and by working under conditions of a task-oriented model of care [13]. Young physicians named problems balancing personal needs with the demands of the workplace to meet the needs of patients during the pandemic [17]. Mental health HCWs reported having additional responsibility in times of increased workload [23]. Further, moral stressors at the organizational level existed in the conflict between professional obligations and family roles [13], reported as an anticipated risk of infecting family members [25]. In this context, fears of infecting one’s family; dilemmas between the desire to help one’s family and the duty to help patients; and the effect of COVID-19 on personal relationships in the form of fear of infecting others were reported as moral stressors [15]. 

#### 3.3.2. Moral Distress and Moral Injury during COVID-19

Three studies measured moral distress as the frequency and intensity of moral stressors [14,16,25]. In one study, MD was measured by the frequency of exposure to moral stressors [15]. Based on these operationalizations used in the quantitative studies, MD was moderate [15,16,25] or low [14]. However, in Norman and colleagues [15] 53–88% of HCWs reported having experienced moral distress and in Wilson and colleagues [20], respondents experienced MD 2–3 times a week [20]. Referring to the prevalence of MI in HCWs, three quantitative studies assessed the psychological impact of morally distressing events by using the MISS-HP [11]. Different results emerged; for example, in a Chinese study, 41.3% of respondents reported MI and 20.4% clinically relevant MI [10]. In an Israeli study, 41% of health workers reported clinically relevant symptomatology of MI [79]. In a study from the USA, 23.9% of HCWs reported at least moderate symptoms of MI and 7.8% stated clinically relevant MI [11]. Here the clinical syndrome [28,36] of MI included the dimensions of betrayal, guilt, shame, moral concerns, loss of trust, loss of meaning, difficulty to forgive, and self-condemnation. Additional criteria represented religious struggle and loss of religious faith [11]. 

Symptoms of MI were guilt [22,23] sadness [1], anxiety [22], helplessness [1], loss of confidence [22], and isolation [1]. They were often the result of individual stressors whereas blame, frustration [17,19,22], cynicism [17], and anger [19] were triggered in the context of other related stressors at the team or organizational level. A lack of trust in leadership, loss of trust, and diminished commitment toward the organization were reported as outcomes [19].

#### 3.3.3. Vulnerability Indicators for Moral Distress and Moral Injury in Healthcare Workers during COVID-19

Referring to individual risk factors for developing clinically relevant symptoms of MI, no religious affiliation and low identification with religion [10] were identified as vulnerability factors. Further, employees with lower scores in self-compassion and higher scores in self-criticism were more likely to experience betrayal [79]. Age represented a protective factor in one study with reference to exposure to morally distressing events [79]. In line with this, another study positively correlated younger age and less work experience with the occurrence of MI symptoms [11]. In one study, female gender and lower educational background represented vulnerability factors [10].

Referring to the organizational setting, different employment groups reported different rates of exposure to morally distressing events and their consequences. Here, mental health workers, managers, and re-employed workers reported more frequent dilemmas [14]. Nurses reported MI more frequently than psychiatrists [10], probably due to higher exposure to PMIEs [79]. Employees exposed to medical violence by working experiences of physical or verbal violence from patients or relatives were more likely to report MI [10]. With reference to exposure to unsafe working conditions, people working with COVID-19 patients were more likely to develop symptoms of MI than those without contact with COVID-19 patients [10]. Levels of stress and lack of workplace support were positively associated with MI [21]. Ineffective communication was associated with MD in nurses [7,25]. Further, lacking knowledge and experience in triage measures [7,15,23] represented a vulnerability factor.

## 4. Discussion

This review, first, aimed to provide an overview of moral stressors and MI in HCWs during COVID-19. We further aimed to distinguish between causes, namely moral stressors and PMIEs, and consequences, such as MD and MI, in HCWs, in the context of the COVID-19 crisis.

### 4.1. Moral Stressors during COVID-19

Most studies referred to MD by evaluating situations including moral stressors. The classification of morally distressing situations is largely consistent with the characterization of morally distressing events in the nursing literature of Epstein et al. [29]. For example, HCWs during COVID-19 had to carry out their activities under the extreme conditions of the pandemic, in which the possibilities to influence characteristics of the situation were low. These conditions are characterized as moral stressors at individual, social, and organizational levels [29,30]. At the patient care level, there are several forms of moral stressors during COVID-19: dilemmas between patient care and protection from infection for HCWs and family, inadequate patient care, and the conflict between necessary isolation and patients’ freedom [9,13]. Moral stressors at the interpersonal level represent conflicts within teams triggered by changing teams [7,9] and a lack of competence among colleagues [7,15]. Diverging opinions on treatment planning [7] and colleagues not acting according to safety standards [12] represent further interpersonal stressors. Organizational stressors have become apparent in terms of scarce resources of PPE [13,14,25], time [12], and personnel [16]. COVID-19 related moral stressors such as conflicts between personal and family roles and the HCW role [9,13,17,22,23], a lack of PPE [13,14,22,23,25], inadequate knowledge [7,15,23], and crisis contexts, are proposed to be stressors that are not normal, potentially leading to moral injury (expressed often by the use of the term PMIE) [36]. According to Litz and Kerig [28], these distressing events are rare in their occurrence, considering the pandemic context, as well as extreme in the sense of threatening the moral integrity of most people. These events deeply violate the understanding of shared expectations and values among HCWs, patients, and relatives. Central moral conflicts between ethical principles and decision-making are subsequently present as PMIEs with regard to curtailing visitation rights and caring for dying patients [7,9,16,25]. Studies of MI mostly refer to MD as an institutional healthcare specific phenomenon caused by stressors related to values concerning the role of a healthcare professional [12,15,16,17,29]. However MI is said to be the consequence of the violation of personal beliefs and expectations [36,50]. In the context of COVID-19, the border between these definitions seems to blur as decisions about protective equipment and vaccination may involve both personal values and expectations as well as professional expectations of HCWs to protect and ensure patient well-being. Other central values that do not refer exclusively to the professional roles of HCWs include the desire for a dignified death. This complicates the theoretical distinction between personal and culturally universal values [44] and profession-specific values [48,49].

### 4.2. Moral Distress and Moral Injury during COVID-19

Moral stressors may trigger MD. However, if these stressors are not experienced often and continuously, they may be attributed by individuals to situations or circumstances; cognitively processed; and, thus, possess only moderate psychosocial consequences. In these cases, moral distress has no long-term impact on functioning levels [36]. Jameton [30] refers to this short-term response to moral stress as initial moral distress. The relative normativity of experiencing moral stressors [28] is in accordance with the fact that some of the COVID-19 studies report low and medium levels of MD [14,15,16,25], and by this, they do not differ from results of pre-COVID-19 studies in HCWs [32,61]. Nevertheless, many HCWs have continuously experienced morally stressful situations during the pandemic [15,20]. We consider this comparable to cumulative traumatization. Therefore, we suggest that due to the long duration and ubiquity of the COVID-19 pandemic, the distinction between “normal” everyday moral stressors and rare potentially morally injurious events cannot be maintained. The origin of the term PMIE stems from war studies, where traumatic events are more common. In pandemics, the typical MI stems from the continuous experience of many cumulative moral stressors related to either limitations and orders of legitimate authorities or witnessing transgressions of trusted individuals of the organization or community [42,56]. Being confronted with persisting stressors, the experience of MD may accumulate (the crescendo effect) and the moral integrity of the person as a whole, the moral integrity and/or the trustworthiness of the health care system, or of one’s own organization may be severely challenged [36,46,50]. According to Epstein and Hamric [50], this experience is referred to as moral residue. According to Litz and colleagues [36], severe consequences for individual and social wellbeing is referred to as moral injury. 

### 4.3. Consequences of Moral Distress and Moral Injury

In the present studies, MI is associated with low well-being [10,20] and, with symptoms of posttraumatic stress disorder [79] and other psychological problems such as burnout [10,11,20]. Symptoms and comorbidities of MD include depression [79] and low self-compassion [77]. The association with depression is consistent with findings in military personnel [52,53,54]. One longitudinal study identified burnout as a predictor of MD [20]. This implies that persons with preexisting psychological problems are more vulnerable to MD than others. MD and MI do not only have implications for individual well-being and mental health but they have consequences on an organizational level. In the social occupational context, feelings of exhaustion [22], burnout [10,11,20], disengagement from work, and the desire to change career direction [17] are correlates of MI and MD during COVID-19. These factors should be further addressed in longitudinal designs as long-term consequences of persisting MD and MI. Another proposed focus for future research is the conceptualization of MD and MI with reference to burnout. Burnout due to chronic occupational stress shares conceptual parallels with MI and MD [69,80]. These concepts should be addressed in detail in future research with reference to origin, emergence, and psychosocial consequences such as emotional exhaustion and depersonalization, topics that exceed the scope of the current review.

### 4.4. Vulnerability Indicators for Moral Distress and Moral Injury in Healthcare Workers during COVID-19

Different vulnerability factors for MD and MI in the time of COVID-19 can be defined in the reviewed studies. Risk factors for MD and MI during the pandemic may lie in personal or organizational conditions. Individual vulnerability to experiencing moral stressors differs from pre-COVID-19 studies: one individual risk factor for developing MI during COVID-19 appears to be a younger age [10,11], which contradicts former results on age-effects [33,34,57]. Pre-COVID-19 studies report higher age and higher educational background as vulnerability factors [33,57]. One of the COVID-19 studies indicated a lower level of experience as a vulnerability factor [11]. A possible explanation of different effects of morally distressing events may lie in different capacities to cope with moral stressors either through comprehensibility and meaningfulness or by applying problem-solving behaviors [60]. Oh and Gastmans [59] hypothesize that with growing working experience, cumulative moral trauma may either result in learning from morally stressful events or in cumulative MD or traumatization. Both cognitive (meaning-making) [53] and emotional components [51] of the self-referential emotions of guilt and shame play an important role in experiencing MI [51]. Referring to the experience of MI, studies on MI also address the emotion of guilt [22,23]. However, further research into self-referential emotions is needed, especially the emotion of shame. In contrast to guilt, the emotion of shame, or “inappropriate guilt”, leads to fundamental self-censoring [51] and withdrawal from the community [36]. Taking into account the circularity of the emotions of shame, guilt, and blame and their social consequences, both their theoretical embeddedness into a broad model as well as practical considerations in the form of support through community [14,23,51] or informal support [55,81] are important topics for discussion in research and practice.

Organizational vulnerabilities displayed by the reviewed studies represent instrumental leadership [10,22,25], a lack of resources [22,25], new tasks and roles [7,20,22,23], and a lack of communication [7,25]. A lack of leadership support was for example visible in task-orientated functional leadership that makes little individual reference to clients and staff [62]. Further, medical violence, including former exposure to verbal or physical violence of patients or relatives [20], represented a risk factor for developing MI. This result aligns with the distressing effects of poor ethical climate as well as confirming previous trauma exposure as a risk factor for MD [69,70]. Further, lacking organizational support was present in the form of exposure to unsafe working conditions [10] via extreme exposure to the virus. This was based on a lack of PPE [22,25] and because of working in a COVID-19 ward [10] or an intensive care unit [18]. Psychological unpreparedness for morally stressful situations [3] was present for mental health workers, managers, and re-employed workers who increasingly reported moral dilemmas [10,14]. The vulnerability factor of working in unfamiliar roles was supported by Donkers and colleagues [16]. In their study, HCWs working in intensive care units reported less MD than HCWs on other wards [16]. During COVID-19, being confronted with new tasks and roles in relation to critical care and triage decisions [7,20,23] represented a vulnerability factor for MD, especially as mediated by poor communication [25]. Poor communication and cooperation within as well as between the different occupational groups is a known vulnerability factor in healthcare contexts [5,62,66,68]. 

### 4.5. Strengths and Limitations

Strengths and limitations of the present review must be considered. One strength of the present study is the inclusion of qualitative and quantitative studies, leading to a detailed broad picture of moral stressors and consequences during COVID-19. 

Limitations also include the narrow timeframe of 2020 to 2022 from which articles were reviewed. Moral stressors may change as the pandemic prolongs and the extreme events during this crisis may persevere even after the COVID-19 pandemic is overcome. Another limitation is the automatic restriction of the language of articles to English and German. Further, the number of quantitative studies is small and only four of the studies used longitudinal designs that enable making causal claims [11,12,20,21]. The reviewed studies utilized various perspectives on definitions, aetiology, and measurements of MD and MI. Some articles referred to MD [7,9,12,14,15,16,25] using the construct of Litz and colleagues [36] for MI [10,11,13,18,19,21,79] while some referred more to the construct of Shay [42] concerning the betrayal of authorities [22]. Some studies theoretically referred to all of the definitions [1,17]. 

The same problem can be seen in the measurement of the consequences of moral stressors. The absence of consensus on measurement limits the scope of the quantitative articles included in the review. Further, the methods of measurement in the quantitative studies are often not revealed in detail. This should be considered critically, as methods used to assess MD [11,29,75,76,77,78] differ greatly ranging from one item questionnaires [14,20] to the measurement of different subscales [12,16], to self-developed items [13,15] and a COVID-19 specific distress scale [25]. Additionally, measuring tools for PMIEs [18,79] and MI [10,11] have rarely been applied to HCWs. 

Addressing the diverse group of HCWs, it has to be noted that the included studies focused predominantly on nurses and physicians from different departments [7,9,10,11,12,17,18,21,25,79]. Some studies included health associate professionals [13,14,16,20,79]. The study of Norman and colleagues [15] included chaplains as a group of traditional faith healers. Especially with reference to the qualitative studies that included other mental health HCWs such as psychologists or psychotherapists [1,22,23] and referring to higher levels of MD in nurses and physicians working in psychiatry [14], the experiences of chaplains remains underrepresented. For example, in the context of providing support for veterans chaplains play an important role in providing spiritual and emotional support at an informal level [81] but they remain underrepresented in the context of experiencing MI themselves [81]. Finally, there is a lack of studies on health management and support workers. One qualitative study refers to one employment specialist and one paramedic [19]. This issue reveals the importance in investigating other occupational groups within the healthcare context and the need to bring attention to these groups.

## 5. Conclusions

This scoping review highlighted moral stressors for HCWs during COVID-19. To our knowledge, a distinction between moral stressors and PMIEs as causes and moral distress and MI as consequences has not previously been made for healthcare settings [80]. Referring to causes, this review proposes COVID-19 specific moral stressors, such as increased exposure to the virus due to a lack of PPE or a failure to guarantee patients a dignified death. The assumptions regarding the crescendo effect and a moral residue can make a valuable contribution in terms of system inherent factors in explaining MI. Especially in the context of the enduring pandemic, HCWs are at risk of experiencing cumulative moral stressors at social or institutional levels, which may erode trust in authority [42]. For both MD and MI, there is a need for conceptualizing the experience of the psychosocial impact of moral stressors of any degree [36]. The development of a common theoretical model is desirable. Additionally, referring to the devastating psychological and social impacts of MI, further research on risk and protective factors at the individual, social, and community levels is needed. 

## Figures and Tables

**Figure 1 ijerph-19-01666-f001:**
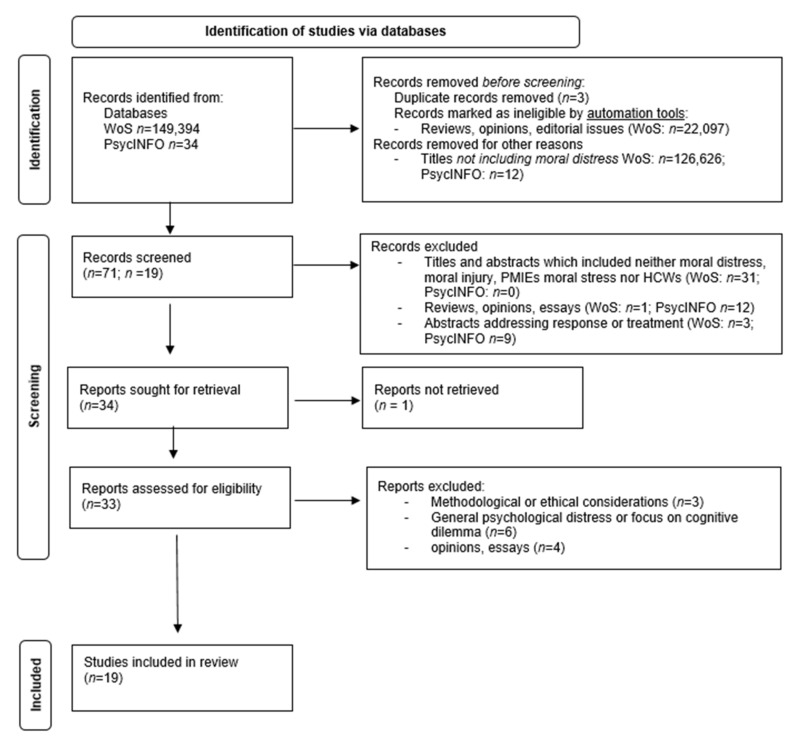
Study selection process.

**Table 1 ijerph-19-01666-t001:** Summary of results.

**Authors (Year)**	**Country**	**Time of Measurement**	**Study Design**	**Sample**	**Measures**	**Main Results**
Wang et al., (2021)	China	March to April 2020	Validation study	3006 doctors and nurses	Moral Injury Symptoms Scale-Health Professional (MISS-HP; [11])	Scores of MISS-HP were positively correlated with depression (r = 0.44), anxiety (r = 0.41), low well-being(r = −0.50), and emotional exhaustion (r = 0.41); 41% of HCWs experience MI.
Mantri et al., (2020)	USA	November 2019 and March 2020	Validation study	181 HCWs (doctors, nurses and “other”)	MISS-HP [11]	Validation of dimensions betrayal, guilt, shame, moral concerns, religious struggle, loss of religious/spiritual faith, loss of meaning/purpose, difficulty forgiving, loss of trust, and self-condemnation as components of MI in HCWs; internal reliability was at 0.75. Discriminant validity was shown by moderate positive correlation of scores with low religiosity, depression, and anxiety symptoms (r’s = 0.25–0.37). Convergent validity was indicated by strong correlations with burnout (*r* = 0.57).
Kok et al., (2021)	Netherlands	October to December 2019 and May to June 2020	Quantitative longitudinal study	233 physicians and nurses in intensive care units of two different hospitals	Moral distress scale-revised (MDS-R; [75])	Differences in the presence of moral stressors before and during COVID-19 prevalent in the context of COVID-19 were hindered care due to a lack of financial support, resources of time or staff; working with colleagues not following safety guidelines or acting unsafely; and working with doctors or nurses who lacked professional competence.
Smallwood et al., (2021)	Australia	August to October 2020	Quantitative study	7846 HCWs, nurses, doctors, and allied health workers	Four self-developed items	Moral distress due to family exclusion; resource constraints; fear of abandoning colleagues in the wake of their own infection; and wearing personal protective equipment (PPE).
Miljeteig et al., (2021)	Norway	April to May 2020	Quantitative study	1606 nurses, managers, and doctors	Moral distress thermometer (MDT; [76] )	Moral distress due to priority setting dilemmas and resource shortages.
Norman et al., (2021)	USA	Spring 2020	Quantitative study	2579 frontline HCWs (physicians, nurse social workers, physician assistants, pastors, and dietitians)	Self-developed 11 Item scale	Moral stressors were present in fears of infecting one’s family; dilemmas between the desire to help one’s family and the duty to help patients; and the effect of COVID-19 on personal relationships and work-related concerns.
Donkers et al., (2021)	Netherlands	April and June 2020	Quantitative study	84 intensive care units in the Netherlands including 355 nurses, 40 intensivists, and 103 supporting staff	Measure of Moral Distress for Healthcare Professionals (MMD-HP; [29])	Experiences of stress for all groups of HCWs included the inability to provide emotional support to patients when they or their relatives were anxious and stressed as well as the inability to provide a dignified death for patients’ relatives. MD scores during COVID-19 were lower for ICU nurses and intensivists compared to one year before COVID-19.
Lake et al., (2021)	USA	September 2020	Quantitative study	307 caregivers	COVID-19 Moral Distress Scale [25]	A lack of protective equipment and the anticipated risk of infecting family members were identified as moral stressors. MD in nurses was negatively associated with effective communication and availability of protective materials and positively associated with number of COVID-19 patients.
Lui et al., (2021)	China	Post deployment to working in Wuhan with COVID-19 patients	Qualitative study	10 nurses working with COVID 19- patients	Semi-structured interviews	Ethical dilemmas were revealed at the level of clinical care, interpersonal relationships, and care management.
Silverman et al., (2021)	USA	April to May 2020	Qualitative study	31 critical care nurses caring for COVID-19 patients	Focus groups and in-depth interviews	Moral stressors were mentioned in terms of lack of knowledge and uncertainty regarding the novel virus; being overwhelmed by COVID disease; and a fear of the virus leading to suboptimal care.
Patterson et al., (2021)	USA	May and July 2020	Qualitative study	34 primary care clinicians	Informal questionnaire	Problems balancing personal needs with the demands of the workplace to meet the needs of patients. Feelings of helplessness, cynicism, disengagement from work, and a desire to change career direction were stated as PMIEs.
Liberati et al., (2021)	England	June and August 2020	Qualitative study	35 mental health care workers (psychiatrists, nurses, caregivers, psychotherapists, and clinical psychologists)	Semi-structured interviews	Dilemmas existed in clinical decision-making, priority setting, care decisions, trade-offs in therapy delivery and role performance, balancing human contact needs, and infection control as well as low organizational support. Psychosocial consequences included sadness, helplessness, isolation, distress, and burnout.
Maftei & Holman, (2021)	Romania	April 2020	Quantitative study	114 doctors	Adopted version of the Moral Injury Events Scale (MIES; [77])	47% of respondents reported high exposure to PMIEs. No associations between PMIE exposure, demographic characteristics or workplace environment (COVID-19 or non-COVID-19) were found. Exposure to PMIEs was associated with physical and emotional impacts.
Zerach & Levi-Belz, (2021)	Israel	February to March 2021	Quantitative study	296 Israeli social workers and hospital staff	MIES; [77] and MISS-HP; [11]	55% reported being betrayed by their leadership, 46% felt they witnessed things that were morally wrong, 32% felt betrayed by people outside the hospital, 32% reported their own moral transgressions, 49% reported having experienced at least one transgression by others, and 62% had experienced betrayal by others. “High Exposure” and “betrayal-only” classes show higher levels of depressive, anxiety, posttraumatic, and more moral injury symptoms compared to the “minimal exposure” class. “High exposure” and “betrayal-only” classes state lower levels of self-compassion and higher levels of self-criticism, relative to participants in the “minimal exposure” class.
French, Hanna, & Huckle, (2021)	England	No date	Qualitative study	16 HCWs (nurses, doctors, therapists, paramedics, head of nursing)	Interviews	Respondents reported experiences of betrayal by management during COVID-19. Staff lacked management support, perceived treatment during the pandemic as dehumanizing, and reported being treated as a replaceable resource. Employees reported a lack of empathy, appreciation, and respect from supervisors; and emotions of frustration, anger, and loss of trust.
Kreh et al., (2021)	Italy and Austria	March to May 2020	Qualitative study	13 key informants (doctors, nurses, psychologists in leading positions)	Interviews	Moral Injury (MI) was represented by feelings of anxiety, blame, frustration, loss of confidence, and exhaustion.
Billings et al., (2021)	England	July 2020	Qualitative study	28 mental health workers from different settings	Interviews	Identification of PMIEs in additional responsibility and increased workload; confidence building with limited visibility of facial expressions due to PPE; isolation, insecurities, and fears due to lack of knowledge; inconsistency in delivery of own services; and blurred roles occurring when colleagues became clients as PMIEs. Identification of MI in feelings of guilt towards patients.
Wilson et al., (2021)	6 countries (not specified)	April and December 2020	Quantitative longitudinal study	378 HCWs (massage therapists, nurses, physicians, and other healthcare personnel)	Single-item Moral Distress Questionnaire [78]	Negative association of MD with mental health and MD was found as a predictor of burnout.
Hines et al., (2021)	USA	March to July 2020	Quantitative longitudinal study	77 critical care staff (90% physicians)	MIES [77])	A supportive workplace environment was associated with low MI; and stressful and less supportive working conditions were associated higher MI.

## Data Availability

The data presented in this study are available within the article. The review protocol can be found in the Appendix A.

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
