# Peer review of "A Scoping Review of Moral Stressors, Moral Distress and Moral Injury in Healthcare Workers during COVID-19"

_ijerph, 2022, doi:10.3390/ijerph19031666_

Round 1

Reviewer 1 Report

Intriguing paper that brings together significant parts of the MI/MD literature in healthcare.

Better clarify MD/MI definitions and differences. Most MD literature notes the violation must be of a professional ethical/institutional nature, not personally held beliefs. Most MI definitions allow for personal values/beliefs as well as ethical/institutional.

Most of the references in the introduction relate to nurses, and only several to physicians. The article title references healthcare workers, are the authors able to include references to more disciplines?

The review identified some references to burnout, could the authors clarify why they chose to exclude burnout theoretically from their summary of the existing literature in the introduction and/or the methods section? It isn’t a problem that it is a excluded, but I think it would be helpful to mention why.

More needs to be added to the limitations section of the paper. Most (if not all) of the studies identified in the review were limited in their ability to determine causality. State this clearly and succinctly. Measurement, definition, and the etiology of MI/MD (even burnout for that matter) are still subject to much debate. No agreement on a “gold standard” measure for each exist. Again, state this clearly as a limitation.

Portion of the MI/PTSD literature in military/veterans distinguish or reference differences in "guilt" and "inappropriate guilt". Did the authors notice any distinctions on their review? See below:

Currier, J. M., Holland, J. M., & Malott, J. (2015). Moral injury, meaning making, and mental health in returning veterans. Journal of clinical psychology, 71(3), 229-240.   Harris, J. I., Park, C. L., Currier, J. M., Usset, T. J., & Voecks, C. D. (2015). Moral injury and psycho-spiritual development: Considering the developmental context. Spirituality in Clinical Practice, 2(4), 256.   Farnsworth, J. K., Drescher, K. D., Nieuwsma, J. A., Walser, R. B., & Currier, J. M. (2014). The role of moral emotions in military trauma: Implications for the study and treatment of moral injury. Review of General Psychology, 18(4), 249-262.   Additions of more paragraphs to break up the text would be helpful (especially in 1.2/1.3 and the discussion) for ease of reading. 

Thank you!

Author Response

We thank the reviewer for the detailed and helpful comments. They assisted us in revising our manuscript and we hope that we significantly improved it. We respond to each comment below and explain how we revised the manuscript.

Intriguing paper that brings together significant parts of the MI/MD literature in healthcare. Better clarify MD/MI definitions and differences. Most MD literature notes the violation must be of a professional ethical/institutional nature, not personally held beliefs. Most MI definitions allow for personal values/beliefs as well as ethical/institutional.

Thank you for your helpful comment. By adding some parts into the theoretical section, we hopefully specified more clearly between MD and MI. With reference to MD, we added references regarding values in the healthcare profession (Lines 109-113). To address MI, we highlighted Shay’s perspective (Lines 168-173), which emphasizes the social component. Additionally we included the perspective of Jinkerson (2016), who stresses social and individual components in his Syndrome perspective of MI. The perspective of Carey & Hodgson (2018) embedding MI into a bio-psycho-social-spiritual model may help provide context regarding emotions in the context of MI, and was mentioned as well (Lines 174-176).

As both healthcare values and personal values are mostly the same, e.g., reduction of harm or dignified death or grief, we discussed the issue of the differentiation of MI and MD, and made brief remarks concerning this issue in the discussion section (Lines 144-151).

Most of the references in the introduction relate to nurses, and only several to physicians. The article title references healthcare workers, are the authors able to include references to more disciplines?

Thank you for this important question. It is an unfortunate issue that most of the research on MD focusses on nurses and more recently on physicians in hospitals. Though our search strategy included HCWs, the reviewed studies display a huge knowledge gap in related professions for example in (paramedics/firefighters) or social workers, and in chaplains. We added a more explicit discussion of this gap in studies involving representatives of other professional groups into our limitation section (Lines 262-275).

The review identified some references to burnout, could the authors clarify why they chose to exclude burnout theoretically from their summary of the existing literature in the introduction and/or the methods section? It isn’t a problem that it is a excluded, but I think it would be helpful to mention why.

Thank you for your question. We agree that burnout is an important aspect, especially in healthcare. Additionally, concepts of MD and Mi have several etiological parallels to the syndrome of burnout. We excluded burnout from our literature review to limit the length of our article and to limit our scope to moral stressors and MI and MD. To remark on the importance of the construct of burnout, we added a statement on the need for additional research on burnout and MI/MD in the discussion section (Lines 188-193)

More needs to be added to the limitations section of the paper. Most (if not all) of the studies identified in the review were limited in their ability to determine causality. State this clearly and succinctly. Measurement, definition, and the etiology of MI/MD (even burnout for that matter) are still subject to much debate. No agreement on a “gold standard” measure for each exist. Again, state this clearly as a limitation.

Thank you for this remark. We agree that the limitation section lacked several points. We improved the limitation section. Specifically, we remarked on the important issue of a lack of consensus regarding measurement, on the absence of drawing causality in most of the studies, as well as on the usage of different operationalisations in the reviewed studies. As we considered your remark of definition and etiology of burnout and MI/MD to be especially important, we added additional discussion about this issue (Lines 240-261).

Portion of the MI/PTSD literature in military/veterans distinguish or reference differences in "guilt" and "inappropriate guilt". Did the authors notice any distinctions on their review? See below:

Currier, J. M., Holland, J. M., & Malott, J. (2015). Moral injury, meaning making, and mental health in returning veterans. Journal of clinical psychology, 71(3), 229-240. Harris, J. I., Park, C. L., Currier, J. M., Usset, T. J., & Voecks, C. D. (2015). Moral injury and psycho-spiritual development: Considering the developmental context. Spirituality in Clinical Practice, 2(4), 256.

Farnsworth, J. K., Drescher, K. D., Nieuwsma, J. A., Walser, R. B., & Currier, J. M. (2014). The role of moral emotions in military trauma: Implications for the study and treatment of moral injury. Review of General Psychology, 18(4), 249-262.

Thank you for this central question. We confirm that moral emotions, especially potentially self-harming emotions of “inappropriate guilt”, are important in the context of MI. There is one qualitative article (Line 187) that states guilt as a “symptom” in mental HCWs, but none which differentiates guilt and inappropriate guilt. Most papers are less focused on the consequences of stressors than on the causes. As moral emotions are components of MI and mostly not included in the studies, we remarked on the importance of investigating moral emotions and cognitive factors such as meaning-making as possible factors influencing the emergence and maintenance of MI (Lines 208-214).

Additions of more paragraphs to break up the text would be helpful (especially in 1.2/1.3 and the discussion) for ease of reading.

Thank you for this remark! We added more paragraphs in the theoretical section to make it easier to read.

Reviewer 2 Report

This is an important paper and should be published eventually but not until an extensive editing takes place. There are numerous typos, grammatical errors, and awkward sentences that distract from the quality of the research. Specifically, the following lines have grammatical issues:

page 2:

lines - 55-56, 63, 70, 78, 87-91

page 3:

lines - 99-100, 101, 103, 106, 115, 131, 143

page 4:

lines - 158, 174, 179, 193, 194

page 5:

lines- 235-237

page 12:

lines 4-14

page 14

lines- 118, 125-127, 130, 140, 147, 155, 160

page 15:

lines- 181, 182, 186, 193, 194, 202

Author Response

This is an important paper and should be published eventually but not until an extensive editing takes place. There are numerous typos, grammatical errors, and awkward sentences that distract from the quality of the research. Specifically, the following lines have grammatical issues:

Thank you for the feedback, we recognize that linguistic revisions were needed. We have made a full linguistic revision with particular attention to the areas mentioned by the reviewer. We hope that the language in the manuscript is now comprehensive and meets editorial expectations.

Reviewer 3 Report

This is a well-written and presented paper. Congratulations to the authors.  However, my points of contention with this paper is that it repeatedly notes HCWs but nowhere does it detail who/what comprises a HCW; Yes it suggests doctors, nurses, psychologists, social workers MHW - but who else? (e.g., physical therapists, occupational therapists, audiologists, speech-language pathologists, etc) - surely all HCWs are relevant, but it seems to be very vague about this in the definition, nor qualifies clearly which HCWs are involved in what literature (see Table 1). I would have thought this fundamentally important since the thrust is about HCWs.

Along with the above (though I realize it will probably not alter the results), however, the authors do note one of the HCWs being 'clergy'; but firstly I would edit this to be clergy/chaplains (given chaplaincy terminology being more commonly associated with HCWs), but secondly, this paper fails to note the connection or relevance of clergy/chaplains to MI and particularly with regard to the military (p.2) - which has been a main area of research.  I recommend the authors consider the extensive reviews and definitions of MI by Carey, Hodgson et al (2016) and Hodgson & Carey (2017) with respect to chaplains and MI; even if it means the authors conclude that very little was found/excluded with regard to this particular HCW profession given the research aims - but at least it would show evidence of the authors' knowledge of this literature and the consolidated definition for MI utilized by Carey & Hodgson (2018) in Frontiers in Psychiatry.

Thirdly, the focus of this paper seems to reflect a strong bias towards the individual as being the central 'actor' with regards to M-Stress, MD and MI. While this largely aligns with Litz's approach to MI, this emphasis contradicts the work of Shay who clearly takes a 'systems' approach to examine MI and critiques organizational leadership as the predominant cause of MI and subsequent bio-psycho-social-spiritual damage to individuals. Shay's perspective needs to be made clearer in this paper - as his systems perspective seems most relevant with regard to the issues/contentions that HCWs endure with respect to M-Stress, MD and MI.

Author Response

We thank the reviewer for the detailed and helpful comments. They assisted us in revising our manuscript and we hope that we significantly improved it. We respond to each comment below and explain how we revised the manuscript.

This is a well-written and presented paper. Congratulations to the authors. However, my points of contention with this paper is that it repeatedly notes HCWs but nowhere does it detail who/what comprises a HCW; Yes it suggests doctors, nurses, psychologists, social workers MHW - but who else? (e.g., physical therapists, occupational therapists, audiologists, speech-language pathologists, etc) - surely all HCWs are relevant, but it seems to be very vague about this in the definition, nor qualifies clearly which HCWs are involved in what literature (see Table 1). I would have thought this fundamentally important since the thrust is about HCWs.

Thank you for your important remark. By using the definition of the WHO (2006) we tried to frame the general profession of HCWs more explicitly. This definition allows for HCWs to include health service providers delivering personal or non-personal services, and health management and support workers (lines 247-252). Having included HCWs in addition to the dominant group of nurses and doctors in our search strategy, this ended up yielding different types of workers. When referring to HCWs we refer to the group examined in each stated article.

Along with the above (though I realize it will probably not alter the results), however, the authors do note one of the HCWs being 'clergy'; but firstly I would edit this to be clergy/chaplains (given chaplaincy terminology being more commonly associated with HCWs), but secondly, this paper fails to note the connection or relevance of clergy/chaplains to MI and particularly with regard to the military (p.2) - which has been a main area of research. I recommend the authors consider the extensive reviews and definitions of MI by Carey, Hodgson et al (2016) and Hodgson & Carey (2017) with respect to chaplains and MI; even if it means the authors conclude that very little was found/excluded with regard to this particular HCW profession given the research aims - but at least it would show evidence of the authors' knowledge of this literature and the consolidated definition for MI utilized by Carey & Hodgson (2018) in Frontiers in Psychiatry.

Thank you for the helpful comment and for the central literature advice. We changed the terminology by editing clergy into chaplains, which was important to highlighting the relevance of the literature on chaplains in MI literature. Especially with reference to coping, chaplains play an important role. We added language about the important role of chaplains in providing informal support, as well as the problem of the lack of literature on MI in chaplains into our discussion. To stress both the individual and social determinants of MI, we thank the reviewer for the helpful remark on the definition used by Carey & Hodgson (2018) and have integrated it into our work (lines 174-191). In the discussion section, we also draw attention to the need to explore MD and MI in chaplains (lines 268-271).

Thirdly, the focus of this paper seems to reflect a strong bias towards the individual as being the central 'actor' with regards to M-Stress, MD and MI. While this largely aligns with Litz's approach to MI, this emphasis contradicts the work of Shay who clearly takes a 'systems' approach to examine MI and critiques organizational leadership as the predominant cause of MI and subsequent bio-psycho-social-spiritual damage to individuals. Shay's perspective needs to be made clearer in this paper - as his systems perspective seems most relevant with regard to the issues/contentions that HCWs endure with respect to M-Stress, MD and MI.

Thank you for your important comment. Shay’s authority-focused perspective is a very important one. To point this out, we stated more clearly the differences between the perspectives of Litz and collegues (2009) and Shay (2014) (lines 168-183). As we consider the perspectives of Jinkerson (2016) and Carey & Hodgson (2018) to be important in adding the component of spirituality into a holistic model of MI, we reinforced the importance for further research into the bio-psycho-social-spiritual model for MI (lines 235-238).

Round 2

Reviewer 2 Report

The manuscript has been suitably revised and I recommend acceptance.

Author Response

We thank all reviewers who provided feedback; your comments were instrumental to improving our manuscript.